# Single-photon emission from single-electron transport in a SAW-driven lateral light-emitting diode

Tzu-Kan Hsiao [1,3✉], Antonio Rubino[1,4], Yousun Chung[1,5], Seok-Kyun Son [1,6], Hangtian Hou[1], Jorge Pedrós [1,7], Ateeq Nasir [1,2], Gabriel Éthier-Majcher[1], Megan J. Stanley[1,8], Richard T. Phillips [1], Thomas A. Mitchell[1], Jonathan P. Griffiths[1], Ian Farrer [1,9], David A. Ritchie [1] & Christopher J.B. Ford [1✉]

The long-distance quantum transfer between electron-spin qubits in semiconductors is important for realising large-scale quantum computing circuits. Electron-spin to photon-polarisation conversion is a promising technology for achieving free-space or fibre-coupled quantum transfer. In this work, using only regular lithography techniques on a conventional 15 nm GaAs quantum well, we demonstrate acoustically-driven generation of single photons from single electrons, without the need for a self-assembled quantum dot. In this device, a single electron is carried in a potential minimum of a surface acoustic wave (SAW) and is transported to a region of holes to form an exciton. The exciton then decays and creates a single optical photon within 100 ps. This SAW-driven electroluminescence, without optimisation, yields photon antibunching with $g^{(2)}(0) = 0.39 \pm 0.05$ in the single-electron limit ($g^{(2)}(0) = 0.63 \pm 0.03$ in the raw histogram). Our work marks the first step towards electron-to-photon (spin-to-polarisation) qubit conversion for scaleable quantum computing architectures.

[1] Department of Physics, Cavendish Laboratory, University of Cambridge, Cambridge CB3 0HE, UK. [2] National Physical Laboratory, Hampton Road, Teddington TW11 0LW, UK. [3] Present address: QuTech, Delft University of Technology, Delft 2628 CJ, Netherlands. [4] Present address: LS Instruments AG, Passage du Cardinal 1, 1700 Fribourg, Switzerland. [5] Present address: Centre of Excellence for Quantum Computation and Communication Technology, University of New South Wales, Sydney, NSW 2052, Australia. [6] Present address: Department of Physics, Mokpo National University, Muan, Jeollanam-do 58554, Republic of Korea. [7] Present address: Instituto de Sistemas Optoelectrónicos y Microtecnología and Departamento de Ingeniería Electrónica, E.T.S.I de Telecomunicación, Universidad Polit écnica de Madrid, Madrid 28040, Spain. [8] Present address: University of California, San Francisco, 185 Berry Street Bldg B, San Francisco, CA 94158, USA. [9] Present address: Department of Electronic and Electrical Engineering, University of Sheffield, Mappin St, Sheffield S1 3JD, UK. ✉email: tzu-kan.hsiao@cantab.net; cjbf@cam.ac.uk

The long-distance transfer of quantum information encoded as electron-spin qubits in semiconductor quantum dots has been extensively studied because of its importance in realising scaleable quantum-computing architectures[1]. While high-fidelity qubit operations have been achieved in small-scale few-qubit systems[2–4], long-distance coupling between distant spin qubits in a large quantum circuit is a big challenge since the exchange interaction relies on the overlapping of two electron wavefunctions. To solve this issue, many methods such as repeated SWAP operations[5], electron shuttling[6–8], and surface acoustic waves (SAWs)[9–12], have demonstrated quantum information transfer within a 10 μm length scale. Spin-microwave-resonator coupling is also investigated, and the coupling between two electron spins on a millimetre scale has been demonstrated[13].

Semiconductors like GaAs are a good platform for converting electron spins to the polarisations of optical photons. Such an electron-to-photon (spin-to-polarisation) interface is a promising method to achieve kilometre-scale quantum information transfer since the polarisation-encoded qubits are robust to decoherence in free space and optical fibres[14,15]. Hence, large-scale distributed quantum-computing networks may be constructed by combining with the inverse (polarisation-to-spin) conversion[16]. In order to achieve this ambitious goal, the first major step is to develop a technology that is capable of converting a single-electron state to a single-photon state, and is also compatible with conventional semiconductor fabrication for device integration.

A scheme that can be used for producing single photons from single electrons was proposed by Foden et al.[17]. It was originally proposed for making a SAW-driven single-photon source with a high repetition rate. This makes use of the fact that, in a piezoelectric material such as GaAs, a SAW consists of both a strain and a potential modulation. In a narrow channel, electrons are confined in moving quantum dots formed by the SAW potential and the lateral channel potential. The number, $n$, of electrons in each SAW potential minimum is well defined if the Coulomb charging energy is large enough. The SAW (of frequency $f_{SAW}$) can therefore drive a quantised current $nef_{SAW}$ along the channel ($e$ is the electronic charge)[18,19]. To generate photon emission, single electrons must be carried in SAW potential minima across a lateral n-i-p junction to create single photons by recombining with holes (see Supplementary Movie 1 for a schematic animation). Using this scheme, a propagating single-electron state is converted to a single-photon state.

To demonstrate the electron-to-photon conversion in the single-electron limit, the second-order correlation function, $g^{(2)}$(0), where $g^{(2)}(0) \leq 1$ for sub-Poissonian light and $g^{(2)}(0) \leq 0.5$ for single-photon emission, needs to be measured when the SAW-driven current $\sim 1ef_{SAW}$. For two decades, various attempts have been made to implement this scheme but single-photon emission has not been observed[20–25]. Our recent work shows quantised SAW-driven current in a gate-induced n-i-n junction, indicating a promising route for an electron-to-photon interface using gate-induced n-i-p junction[26].

We note that several works based on SAW-injected excitons into a self-assembled quantum dot[27,28] or a SAW-modulated Purcell effect in a dot-cavity system[29,30] have shown single-photon emission. Long-distance transport of optically excited carriers along an etched quantum wire is also achieved using a SAW[31]. These works demonstrate the capability of a SAW to transport and inject single excitons, and to dynamically modulate the coupling between a quantum dot and a cavity, but these results rely on optically excited carriers and the presence of a randomly formed self-assembled quantum dot or recombination centre, which will pose a challenge in device integration.

Here we successfully demonstrate the generation of single photons from single electrons using Foden's scheme. Our device

is fabricated using a deterministic conventional lithography process, which is compatible with gate-defined quantum dots. In the single-electron limit, without any spectral filtering, our device shows clear photon antibunching with $g^{(2)}(0) = 0.39 \pm 0.05$, indicating single-photon emission created by single-electron transport.

## Results

**Device and setup**. In this work, the lateral n-i-p junction is made in a conventional undoped 15 nm GaAs quantum well using standard lithography techniques (see the "Methods" section). Electrons and holes are induced in the regions under the electron and hole surface gates, which are separated by an intrinsic region (Fig. 1a). A source-drain (S-D) bias less than the GaAs bandgap is applied across the n-i-p junction to create a finite potential difference between the electron and hole regions. A SAW is generated by applying a radio-frequency (RF) signal to an interdigitated transducer (IDT) at its resonant frequency $f_{SAW} = 1.163$ GHz. Electrons are trapped in SAW potential minima and pushed towards the hole region (Fig. 1b).

To achieve SAW-driven single-electron transport, lateral confinement is provided by etching the region connecting the electron and hole regions into a 1D intrinsic channel. In addition, a pair of side gates is placed on either side of the channel to adjust the electrostatic potential in the intrinsic region. The physical length of the channel is made to be similar to the SAW wavelength of 2.5 μm. In this case, any current flow will be caused by the SAW carrying electrons up the potential slope linking the conduction band in the regions of electrons and holes, not by the SAW reducing the height of the potential barrier in the intrinsic region at a certain part of its cycle. All measurements were carried out at 1.5 K.

**SAW-driven electron transport and electroluminescence**. In order to test the effect of a SAW on the induced lateral n-i-p junction, a S-D bias, $V_{SD}$, <1.45 V is applied to the junction. This is at least 90 mV below the voltage required to align the conduction band in the $n$ and $p$ regions so that a current can flow at cryogenic temperature if any intermediate barrier is overcome. In this case, due to the conduction-band offset between the $n$ and $p$ regions, electrons cannot reach the $p$ region to recombine with holes unless a SAW carries them there. Therefore, a S-D current and electroluminescence (EL) signal will only appear when an RF signal is applied to the IDT at $f_{SAW}$.

The SAW-driven current and EL are shown in Fig. 1c. The S-D current (Fig. 1c top panel) is greatly enhanced around $f_{SAW} \sim$ 1.163 GHz with an RF power of 9 dBm (quality factor $\frac{f_{SAW}}{\Delta f} \sim 390$). This SAW-driven current is close to 1 $ef_{SAW} = 0.186$ nA. It means that the number of electrons carried in each SAW minimum is roughly one on average, a single-electron regime which will, in principle, generate single photons. These electrons driven by the SAW arrive at the hole region and recombine, causing a SAW-driven EL signal, as seen in Fig. 1c (bottom panel). The EL signal is emitted from the $p$ region as electrons recombine with holes there.

The internal quantum efficiency, $\eta$, defined as the ratio of the number of photons actually collected to the number of photons that can theoretically be collected by the optics, is about 2.5% (see the "Methods" section). This low $\eta$ may be caused by trapping and non-radiative recombination in surface states around the etched edges[32], or due to electrons being carried away without recombining near the junction. The time-resolved measurement of the SAW-driven EL, shown in Fig. 1d, exhibits periodic peaks with a period of 860 ps. Hence, it is evident that electrons are

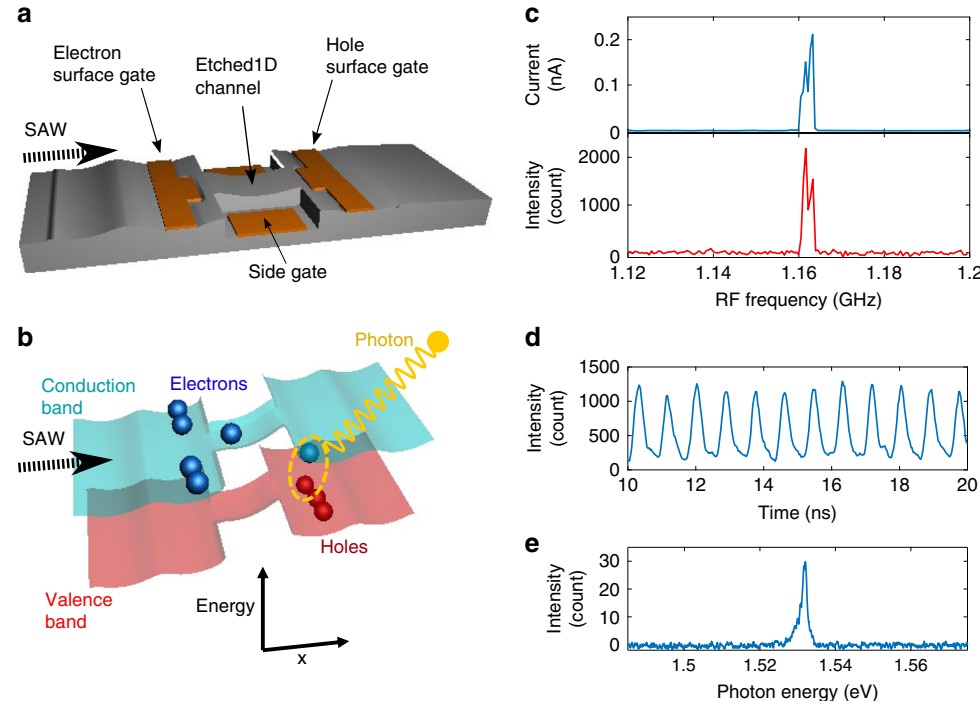

**Fig. 1 SAW-driven lateral *n-i-p* junction, and its electrical and optical properties. a** Schematic of the device. Electron and hole surface gates induce electrons (*n*-region) and holes (*p*-region) in a GaAs quantum well, forming a lateral *n-i-p* junction along an etched 1D channel. A SAW is generated by applying an RF signal to a transducer (placed 1 mm from the *n-i-p* junction). **b** Schematic diagram showing the band structure of the *n-i-p* junction modulated by the SAW potential, for an applied forward bias less than the bandgap. A single electron is carried in each SAW minimum, creating a single photon when it recombines with a hole. **c** S-D current (top) and EL intensity (bottom) as a function of applied RF frequency at an RF power of 9 dBm. They both show a significant enhancement around 1.163 GHz, which is the resonant SAW frequency of the IDT. **d** SAW-driven EL intensity as a function of time. The 860 ps periodic feature corresponds to the applied SAW frequency of 1.163 GHz. **e** Energy spectrum of the SAW-driven EL. The spectrum shows a peak at 1.531 eV (FWHM ~ 1 meV), which matches the exciton energy in the quantum well (see Supplementary Note 1).

injected into the hole region by the SAW, leading to photon emission with the period of the SAW.

The spectrum of the SAW-driven EL is shown in Fig. 1e. The spectral peak corresponds to the neutral-exciton transition from the conduction band to the first heavy-hole subband in the quantum well (see Supplementary Note 1)[33]. The full width at half maximum (FWHM) of the peak is about 1 meV, which can be attributed to acoustic-phonon scattering ($\Delta E \sim 0.2$ meV at 1.5 K) and interface roughness (atomic monolayer fluctuations in the quantum-well thickness give $\Delta E \sim 0.5$ meV)[34]. The lower-energy tail of the peak may be due to localised exciton states or a Stark shift in the hole region. Unlike conventional single-photon emission based on self-assembled quantum dots, which usually have an extra peak in the spectrum due to biexciton states, this device shows only one peak (neutral exciton) without any spectral filtering or optical cavity.

**Time-resolved SAW-driven electroluminescence**. The dynamics of the SAW-driven generation of single photons from single electrons can be studied using a time-resolved EL measurement technique. A 350 ns-long pulsed RF signal is applied to the IDT to generate a pulsed SAW (Fig. 2a top). The SAW-driven current is close to the single-electron regime. Because the SAW velocity on GaAs is ~2800 m/s and the distance from the IDT to the *n-i-p* junction is ~1.1 mm, it will take about 400 ns for the SAW to arrive at the junction, and for its amplitude to build up so that it transports electrons which then recombine with holes. Therefore, compared with the RF signal, the SAW-driven EL is delayed by about 400 ns, as can be seen in Fig 2a (bottom). This confirms that the EL signal is indeed caused by the SAW, rather than by

electromagnetic crosstalk generated by the RF signal, which should have an effect without any noticeable delay since the speed of light is five orders of magnitude faster than the SAW.

In order to understand more detailed dynamics, data points three SAW periods apart are averaged across a large part of the region where the EL signal is observed, in Fig. 2a (bottom), to give three periods that are the combination of every third period of the data. The resulting data is shown in Fig. 2b. The shape of an individual peak can be understood from the injection of electrons by the SAW. When an electron is pumped across the *n-i-p* junction by the SAW, the probability of electron-hole recombination suddenly steps up and causes a rapid enhancement of the EL signal. The signal then decays exponentially as the probability that the electron has already recombined rises. The peaks in Fig. 2b are broadened by the temporal uncertainty (jitter) of the single-photon avalanche photodiode (SPAD) and of the SAW-driven electron transport itself, originating from a slight uncertainty about the position of an electron in a SAW minimum. Note that each peak in Fig. 2b does not decay to zero by the time the next peak appears. The reason for this non-zero background level may be due to after-pulsing of the SPAD[35] or to slowly decaying secondary-exciton states (lifetime ~0.2–1.5 ns)[36,37]. These slowly decaying exciton states may be the localised excitons seen in the small lower-energy tail in Fig. 1e.

Dynamical parameters, including carrier lifetime, $\tau$, background offset, $BG_{EL}$, and jitter, $w$, are quantified by fitting the data to a function, $H(t)$, describing the SAW-driven EL (see Supplementary Note 2). The best fit, plotted along with the data in Fig. 2b, gives $\tau = 94$ ps, $w = 33$ ps, and $BG_{EL} = 7\%$ of the peak height. The short carrier lifetime of 94 ps is likely to be caused by non-radiative recombination at surface states, which

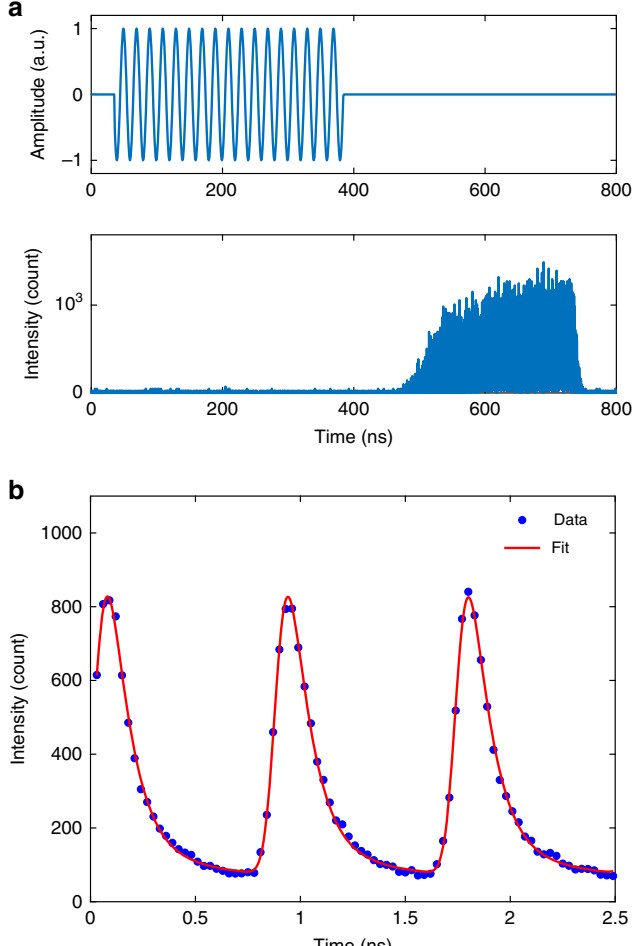

**Fig. 2 Time-resolved measurement of the SAW-driven EL. a** A 350 ns-long pulsed RF signal (top, shown at a low frequency for clarity) is applied to the IDT to create a pulsed SAW. The SAW-driven EL signal (bottom) is delayed by roughly 400 ns owing to propagation of the SAW from the IDT to the $n$-$i$-$p$ junction. **b** Averaged SAW-driven EL and the best fit using $H(t)$ (see Supplementary Note 2).

also gives rise to the observed low quantum efficiency $\eta$. On the other hand, a propagating electron only spends a few hundred ps in the junction area, which would also lead to the observed short lifetime. This carrier lifetime is short compared with the 860 ps SAW period, so photons driven by consecutive SAW minima do not overlap significantly in the time domain.

**Photon antibunching in the single-electron regime.** In this device, quantised SAW-driven current cannot be observed, meaning that there is some variation in the number of electrons in each SAW minimum. However, the probability distribution of electron occupation numbers can still be affected by the discrete nature of SAW-driven charge transport, causing a reduced variance in electron number. The probability distribution should thus become a sub-Poissonian distribution, which will lead to photon antibunching after recombination.

Photon antibunching in the SAW-driven EL is tested by measuring an autocorrelation histogram using a Hanbury Brown and Twiss (HBT) setup (see the "Methods" section). A continuous SAW is used to drive the $n$-$i$-$p$ junction in the single-electron regime (with an average number of electrons in a SAW minimum of 0.89) stabilised by a feedback control loop. Coincidences occurring outside the optimum single-electron

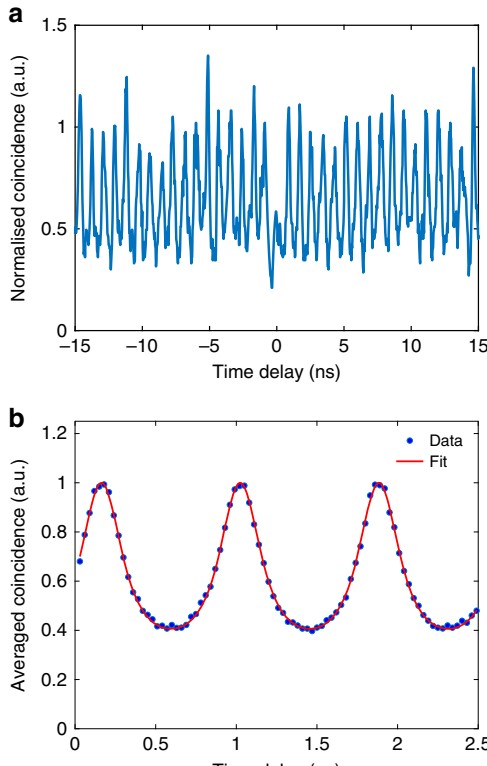

**Fig. 3 Photon antibunching in the SAW-driven EL. a** Normalised autocorrelation histogram of the SAW-driven EL. The coincidence at $\Delta t = 0$ is suppressed to 58% of the average peak value, indicating photon antibunching, i.e., that there is a reduced probability of two photons arriving simultaneously. **b** Averaged autocorrelation histogram and the best fit using $G(\Delta t)$ (see Supplementary Note 3).

regime (SAW-driven current above $1ef$ or below $0.8ef$) are removed from the dataset after acquisition (see the "Methods" section). The autocorrelation histogram as a function of time delay, $\Delta t$, in Fig. 3a shows periodic peaks with the 860 ps SAW period, indicating that coincidences in the histogram are indeed caused by the periodic SAW-driven photon emission. In the single-electron regime, the peak at $\Delta t = 0$ is suppressed to 58% of the average peak value (69% for the raw data, see Supplementary Note 6). The suppression at $\Delta t = 0$ is clear evidence of photon antibunching in the SAW-driven EL (a reduced probability of two photons arriving at the same moment).

**Second-order correlation function.** Although photon antibunching is observed in Fig. 3a, the second-order correlation function $g^{(2)}(\Delta t)$, which confirms the presence of single-photon emission if $g^{(2)}(0) < 0.5$, cannot be simply obtained from the peak heights. This is because coincidence at a peak can have a contribution from the two neighbouring peaks if they have significant overlap, and also because an effective background ($BG_{EL}$) in EL can give rise to a background, $BG_{g2}$, in the autocorrelation histogram. Therefore, the actual shape of individual peaks and the background $BG_{g2}$ have to be considered in order to extract the real $g^{(2)}(\Delta t)$. The peak shape and the background can be estimated by fitting the autocorrelation histogram to a function, $G(\Delta t)$, describing the autocorrelation of SAW-driven EL (see Supplementary Note 3). To have a better fit, points from every third peak in the histogram are averaged together. The averaged histogram and the best fit are plotted in Fig. 3b. The fit shows that the autocorrelation histogram is caused by a SAW-driven EL signal with $\tau = 99$ ps, $w = 33$ ps and $BG_{EL} = 8\%$ of the peak

height. These parameters are consistent with those obtained in fitting the time-resolved data (Fig. 2b). With these parameters, the actual shape of individual peaks and the background $BG_{g2}$ are known. Hence, the real $g^{(2)}(\Delta t)$ can now be extracted from the autocorrelation histogram.

$g^{(2)}(\Delta t)$ of the SAW-driven EL is obtained by finding the real contribution from each peak in the autocorrelation histogram (see Supplementary Note 4). The result is shown in Fig. 4a. In the single-electron regime, the suppressed photon-antibunching peak at $\Delta t = 0$ gives $g^{(2)}(0) = 0.39 \pm 0.05$ ($g^{(2)}(0) = 0.63 \pm 0.03$ for the raw data), showing that the SAW-driven $n$-$i$-$p$ junction can indeed produce single-photon emission from single-electron transport. In addition, since the average number of electrons, $N_{avg}$, in a SAW minimum is 0.89, the probability distribution of photon-number states can be estimated. The wave function of electrons in a SAW minimum can be expressed in the Fock basis

$$|\psi\rangle = \sqrt{P_0}|0\rangle + \sqrt{P_1}|1\rangle + \sqrt{P_2}|2\rangle + \sqrt{P_3}|3\rangle + \cdots$$

where $|n\rangle$ and $P_n$ denote the electron-number states and their respective probabilities. $N_{avg}$ is thus a function of the probability distribution $\{P_n\}$. When $n$ electrons (electron-number state $|n\rangle$) arrive at the hole region, each of these electrons may recombine with a hole and produce a photon according to the internal quantum efficiency $\eta$. Hence, up to $n$ photons are produced from $|n\rangle$. These photons then cause coincidences in an autocorrelation histogram. As a result, $g^{(2)}(0)$ is also a function of the probability distribution $\{P_n\}$. Assuming that $|\psi\rangle$ has no projection on to $|m\rangle$ with $m \geq 4$ (meaning that a SAW minimum can carry only up to

three electrons) and given that $N_{avg} = 0.89$ and $g^{(2)}(0) = 0.39$, the probability distribution $\{P_0, P_1, P_2, P_3\}$ of electron-number states (and photon-number states) in the single-electron regime is estimated to be $\{25 \pm 3\%, 63 \pm 7\%, 10 \pm 6\%, 2 \pm 2\%\}$ (see Supplementary Note 5 for the analysis). This probability distribution is shown in Fig. 4b, along with the distribution expected for a classical Poissonian light source (with the same average number $N_{avg} = 0.89$) for comparison. It can be seen that, in the SAW-driven $n$-$i$-$p$ junction, the single-photon probability is greatly enhanced compared with the classical case. Based on this estimated probability distribution, when a detector receives a light signal from this SAW-driven $n$-$i$-$p$ junction, the signal has a probability of $P_1/(P_1 + P_2 + P_3) = 79–90\%$ to actually be a single photon.

## Discussion

Conversion of single electrons to single photons has been experimentally demonstrated in this SAW-driven lateral $n$-$i$-$p$ junction. However, many improvements need to be made to optimise future devices. In particular, $g^{(2)}(0) = 0.39$ in our work is high compared with $g^{(2)}(0) \sim 1 \times 10^{-4}$ in the very best self-assembled quantum dots[38]. In order to understand how to improve this, we build a simplified SAW-transport model to calculate the probability distribution of SAW-driven electrons. The result indicates that a more well-defined single-electron state (and thus higher $P_1$ and lower $g^{(2)}(0)$) may be achieved with a stronger confinement in the 1D channel and the SAW potential (see Supplementary Note 7). This may be done by using a narrower channel, and depositing ZnO thin film to enhance the SAW potential[39]. As for the low efficiency $\eta$, this can be improved by surface passivation to reduce non-radiative surface states[32], by better capturing of the SAW-driven electrons, and by building an optical cavity (see Supplementary Note 8).

In order to develop spin-to-polarisation qubit conversion, the next step is to generate polarised single photons from single-electron spins by integrating spin-injection techniques with our device[40,41]. In addition, a superposition of spin states can be converted to a superposition of photon polarisations via the recombination of an electron and a light hole under an in-plane magnetic field[42]. Combining the spin-to-polarisation and the inverse polarisation-to-spin conversions[16], one would then achieve long-distance quantum-information transfer between distant semiconductor spin qubits. Moreover, though our experiment is done in GaAs, the same scheme may also be applied to emerging 2D semiconductors, where the spin coherence time $T_2$ is predicted to be about 30 ms in isotopically purified $MoS_2$[43], and the essential SAW-driven charge transport and gate-defined junction have already been realised[44,45]. In addition, as the scheme was originally proposed as a single-photon source, this device may also be useful as a novel single-photon emitter in on-chip quantum photonic networks (see Supplementary Note 9).

In conclusion, we have shown that single photons can be generated using a SAW-driven lateral $n$-$i$-$p$ junction operating in the single-electron limit. This device is fabricated in a fully deterministic lithographic process using gates, etching and an IDT, and hence is compatible with conventional semiconductor fabrication. Such an electron-to-photon conversion interface marks the first major step towards long-distance semiconductor qubit transfer via single optical photons.

## Methods

**Device fabrication**. The SAW-driven lateral $n$-$i$-$p$ junction was made in a 15 nm undoped GaAs quantum well. The quantum-well layer structure consists of (from the top) a 10 nm GaAs capping layer, a 100 nm $Al_{0.33}Ga_{0.67}As$ top barrier, a 15 nm GaAs quantum well, a 285 nm $Al_{0.33}Ga_{0.67}As$ back barrier, and finally a 1 μm GaAs buffer layer. $n$-type and $p$-type ohmic contacts were in direct contact with the

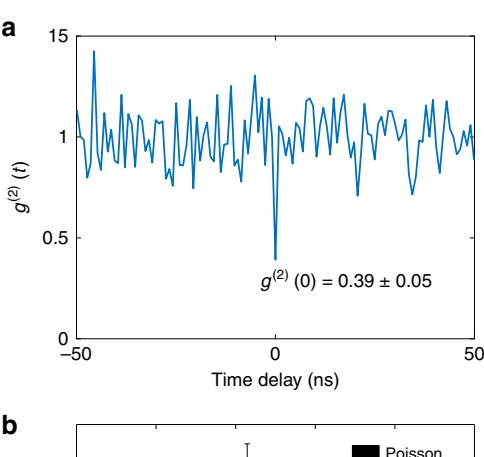

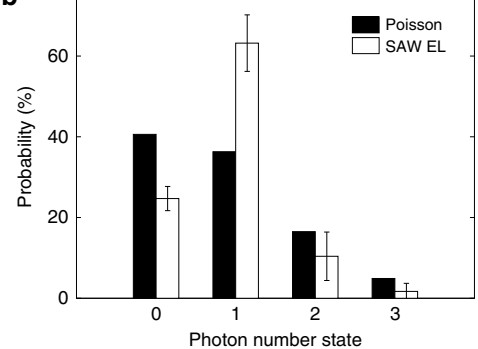

**Fig. 4 SAW-driven single-photon emission in the single-electron regime.** **a** The second-order correlation function $g^{(2)}(\Delta t)$, which is obtained after fitting the autocorrelation histogram to $G'(\Delta t)$. $g^{(2)}(0) = 0.39 \pm 0.05 < 0.5$ shows that the SAW-driven lateral $n$-$i$-$p$ junction produces single-photon emission from the single-electron transport. **b** Estimated probability distribution of photon-number states $|n\rangle$ in the SAW-driven EL, compared with the probability distribution in a Poissonian light source with $N_{avg} = 0.89$. See Supplementary Note 5 for error estimation.

quantum well. AuGeNi (for $n$-type contacts) and AuBe (for $p$-type contacts) were evaporated in regions recessed to the quantum well, and annealed at 470 and 520 °C respectively. Bridging gates for inducing electrons and holes from the ohmic contacts were fabricated by evaporating Ti/Au on a 100 nm $Al_2O_3$ insulator, which is on top of the ohmic contacts. Surface gates for extending charge carriers and forming a lateral $n$-$i$-$p$ junction were fabricated on the wafer surface using electron-beam lithography. 1.2 μm-wide 1D channel between the electron and hole regions was made by removing the quantum well next to the 1D channel using electron-beam-defined wet etching. The IDTs with a period of 2.5 μm were made using electron-beam lithography. The completed device was then mounted on a custom-made sample holder for measurement at 1.5 K.

**Post-selection in the single-electron regime**. Ideally, the device should be operated precisely in the single-electron regime (S-D current = $1ef$). However, even with the S-D bias and all gate voltages kept fixed, the SAW-driven S-D current can drift between $5ef$ (multiple-electron regime) and $0ef$ (vacuum state). This instability may be related to a charging effect near the $n$-$i$-$p$ junction so that the potential drifts and affects the SAW-driven charge transport. In order to deal with this instability, a PID control loop was used so that constant adjustment could be made to S-D bias to keep the SAW-driven current around $0.9ef$. Note that the SAW-driven current still occasionally drifts away from the single-electron regime, so we post-select the coincidences that occurred when the current is between $0.8ef$ and $1.0ef$. We then use these post-selected coincidences (30% of the total coincidences) to analyse $g^{(2)}(0)$ in the single-electron limit. The current measurement had a response time of up to 1 s so sometimes the current might appear to be below 1 $ef$ even though it had drifted up slightly, worsening the statistics. If we use a smaller range of $0.8ef$–$0.9ef$ for the post-selection, $g^{(2)}(0) = 0.38 \pm 0.04$. If a larger range of $0.75ef$–$1.1ef$ is used for selecting coincidences, $g^{(2)}(0)$ becomes $0.47 \pm 0.05$ since it includes more coincidences in the multi-electron regime or vacuum state. Note that even before post-selection, the raw data shows clear photon antibunching with $g^{(2)}(0) = 0.63 \pm 0.03$ for a long measurement period of 54 h. We expect that this charging effect will be reduced by eliminating surface states using surface passivation. Interestingly, this post-selection can be done simultaneously in the measurement because we can know if the device is in the single-electron regime or not by measuring the current in real time rather than looking for photon correlations.

**Optical setup**. An EL signal emitted from a 2 μm$^2$ area is collected by a home-made confocal fibre-coupled lens assembly, the position of which relative to the sample is controlled by a three-axis piezoelectric stage. The EL signal is then sent through a single-mode fibre to optical components outside the cryostat. A 750 mm Czerny-Turner spectrometer with a chilled EMCCD camera is used for taking the EL spectrum. Note that no spectrometer or spectral filtering is involved in the time-resolved EL or HBT experiments.

The internal quantum efficiency $\eta$ is defined as the ratio of the number of photons actually collected to the number of photons that can theoretically be collected by the optics. In the single-electron-transport regime, $\eta$ is determined by

$$\eta = \frac{N_{\text{detect}}}{f_{\text{SAW}} \cdot C_{\text{optics}} \cdot C_{\text{SPAD}}}$$

where $N_{\text{detect}}$ is the number of photons detected by the SPAD per second, $f_{\text{SAW}}$ is the SAW frequency (corresponding to the number of recombinations per second), $C_{\text{optics}}$ is the collection efficiency of the lens assembly (~0.4%), and $C_{\text{SPAD}}$ is the sensitivity of the SPAD at 800 nm (~15%).

**Time-resolved EL measurement setup**. A time-resolved EL setup consists of a SPAD, an arbitrary waveform generator that produces timing pulses, a time-to-digital converter, and a RF source that is synchronised to the timing pulses. The SPAD can be triggered by the detection of a single photon and will then output a signal pulse with a 40 ps jitter. The pulse generator also produces timing pulses with a 10 ps jitter. The signal pulses and the timing pulses are connected to the time-to-digital converter, which measures the time difference between a timing pulse and a signal pulse. The RF source is synchronised with the pulse generator by using a 10 MHz sync signal. The RF signal is used to generate a SAW signal synchronised to the timing pulses. A time-correlated histogram of a SAW-driven EL signal can thus be obtained by recording the time difference between timing pulses and SPAD signal pulses.

**Hanbury Brown-Twiss setup**. An HBT setup consists of a 50:50 fibre-coupled beam-splitter and two SPADs. The beam-splitter splits the stream of photons in a SAW-driven EL signal into two beams. Each beam is sent to a SPAD. These two SPADs produce signal pulses when they are triggered by the incoming photons. In a start-stop autocorrelation method, a signal pulse from SPAD 1 (start) will cause the time-to-digital converter to begin time counting until the counter receives a signal pulse from SPAD 2 (stop). The counter then records one coincidence at the time delay between these two signal pulses. The two beams of photons will give rise to the autocorrelation histogram, which is the number of coincidences as a function of time delay.

## Data availability

Data and processing scripts associated with this work are available at the University of Cambridge data repository (https://doi.org/10.17863/CAM.47728). The source data underlying Figs. 2, 3 and 4 are provided as Source Data files.

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

## Acknowledgements

This work was supported by the European Union Horizon 2020 research and innovation programme under Marie Skłodowska-Curie Grant Agreement No. 642688 (SAWtrain), and the UK EPSRC [Grant Nos. EP/J003417/1 and EP/H017720/1]. A.N. was supported by the UK Department for Business, Innovation and Skills. T.K.H. was supported by the Cambridge Overseas Trust. J.P. acknowledges the support from the Spanish MICINN (Programa Nacional de Movilidad de Recursos Humanos I+D+i) and from the 7th European Community Framework Programme (Marie Curie Intra-European Fellowship, Project no. 235515) during his time at the University of Cambridge, as well as from the Spanish MINECO (grant RyC-2015-18968) now at Universidad Politécnica de Madrid.

We thank Mete Atatüre for the loan of the SPADs and the time-to-digital converter and Charles Smith and Joanna Waldie for the loan of the arbitrary waveform generator. We also thank Peter Spencer for assistance with the optical setup.

## Author contributions

Project planning: C.J.B.F., T.K.H. and J.P.; MBE growth: I.F. and D.A.R.; E-beam lithography: J.P.G. and T.A.M.; sample design and fabrication: T.K.H., Y.C., A.R., S.K.S. and A.N.; electrostatic calculation: H.H.; scanning cryogenic microscope: J.P., Y.C., R.T.P., T.K.H., S.K.S., H.H. and C.J.B.F.; Spectrometer, HBT and time-resolved setup: A.N., G.É.M., M.J.S. and R.T.P.; transport and optical measurements: T.K.H., A.N. and A.R.; analysis of results and theoretical interpretation: T.K.H. and C.J.B.F.

## Competing interests

The authors declare no competing interests.
