## [Peer Review File · Nature Communications]

Reviewers' Comments:

Reviewer #1:

Remarks to the Author:

The authors report on an unconventional type of single photon source. This device consists of an electrostatically induced p-n junction in a two-dimensional quantum well.

Following a proposal dating back to 2000, electrons transported one-by-one via the acoustic-electrical effect into the p-type region, a single photon is emitted.

Single photon emission is verified by performing Hanbury Brown & Twiss intensity interferometry. The observed suppression of correlations at time=0 demonstrates non-classical light emission.

The $g^{(2)}(0)$ value is slightly below 0.5, which is the limit for a perfect source emitting in a single optical mode. This overall performance of the device is still limited but the authors argue that it can be significantly improved by incorporating the structure into an optical cavity to extract light. This reviewer agrees with the authors that this may in fact be feasible.

This reviewer considers the reported research as important to the field of semiconductor quantum electronics because it provides an interface to convert chip-based electron qubits to flying photonic qubits. This reviewer believes that this may be the main application of the demonstrated paradigm and NOT single photon emission. Therefore, this reviewer expects broader impact also in the field of quantum transport and spin-based quantum computing.

This reviewer also recognises that the authors report the first demonstration of this particular type of single photon source.

This reviewer considers the presented work as correct and of high scientific rigour.

The points of novelty listed above warrant in this reviewer's opinion publication in Nature Communications.

Before this reviewer can accept publication, the authors are requested to consider the following two points.

(1) In their manuscript and the motivation of the work the authors set a very strong focus on single photon emission. This field is very advanced, in particular extremely bright, electrically driven sources comprising quantum dots and microresonators are commercially available. Similar holds for room temperature compatible sources using nitrogen vacancy centres in diamond. This reviewer is convinced that the source reported here will not be competitive but find different applications.

In this context this reviewer recommends to improve the manuscript:

The authors cite a long list of works on different types of single photon source (various types of quantum dots, molecules, PDC), several review papers (some on exactly the same topic) and proposals (Quantum internet, linear QCs, etc.) employing single photons.

Work on acoustically driven sources is not made as prominent as it should be. In particular there exists an alternative scheme using quantum dots and the acoustoelectric effect of which only the very first realisation is referenced (ref. 32). There is also a significant body of literature studying the manipulation of optically active quantum emitters, in particular quantum dots via the dynamic strain field of an acoustic wave and regulating the carrier injection by the acoustoelectric effect. This reviewer is convinced the authors are aware of these works and recommends a careful revision such that the reported important results are put in the full context of the field.

(2) The authors mention that the coherence length of their device has to be improved to meet the requirements for their envisioned applications in quantum repeaters. What is the coherence length

of the photons emitted by their current device?

Reviewer #2:

Remarks to the Author:

The authors report on the generation of single photons by injecting (ideally) single carriers using a surface acoustic waves. The full device is fabricated using standard lithography techniques which makes it appealing in terms of reproducibility. The method is interesting, experiments quite well conducted and results are generally well described. I think that the manuscript deserves publication in Nature Communications. Although, there are some major modifications needed before final acceptance.

As a general statement, the results are quite impressive but the authors keep referring to applications and performances they did not investigate (and probably very hard to reach in the future). Since generation of single photons was proven, they should compare with results achieved using electrically driven sources (on various platforms). In several points of the manuscript, they try to convince the reader that their device can generate indistinguishable photons or entangled photons better than other structures (in particular referring to quantum dots). This does not highlight the nice achieved results but forces the reader to compare highly developed performances with arguable theoretical assumptions. This deviates the attention from current data. I suggest the authors to rephrase introduction, conclusion and other parts of the manuscript accordingly. In more details:

1) Quantum dots under optical pumping demonstrated very high degree of indistinguishability (see Ref. [14, 15, 16]) and high extraction efficiency. Additionally, for quantum dots with a lifetime of around 1 ns, linewidth ≤ 1 GHz were reported. The current device generates photons with a bandwidth of 1 meV: the reported lifetime of 100 ps is said to be limited by non-radiative processes; in any case this value is still far from the Fourier limit. This would have a dramatic effect in multiphoton experiments (see H. Wang, et al, Phys. Rev. Lett. 116, 213601 (2016) or J. C. Loredó, et al, Optica 3, 433-440 (2016)).

2) The current results compare very well with electrically driven sources and they would have much better impact if discussed accordingly (see few examples: T. Heindel, et al. New J. Phys. 14, 083001 (2012), J. Zhang, et al, Nat. Commun. 6, 10067 (2015), M. Sartison, et al, Appl. Phys. Lett. 114, 222101 (2019), Z. Yuan, et al, Science 295, 102 (2002), T. Müller, et al, Nat. Commun. 9, 862 (2018))

3) Since the current device produces single photons with a potentially high fabrication yield, I suggest the authors rather than mentioning "Quantum communication and quantum computation", to target quantum cryptography (where only single photons are required).

4) In a similar argument, the generation of polarization entangled photons is well established with quantum dots (see D. Huber, et al, Phys. Rev. Lett. 121, 033902 (2018) or Y. Chen, et al, Nat. Commun. 9, 2994 (2018)). The current device does not need to show this capability and the proposed scheme (polarisation-to-spin conversion) seems quite difficult to implement.

5) In the evaluation of the $g(2)$ function, something seems unclear. The device is operated under pulsed excitation, so that a clear stream of individual peaks can be observed in figure 3 and figure S5. Then a complex procedure to estimate the $g(2)$ is used, obtaining a $g(2)$ function as in figure 4 which looks like a CW second order correlation function. Which new information is given by this figure? Why the authors have to process the data in this way? It is claimed for a better accuracy but adding complex fitting routines should increase the uncertainty. Indeed, the peaks are well separated (despite a slight overlap) and the fitting of figure S5 looks fine. Nevertheless no value of $g(2)$ is reported, while it should be around 0.39. Evaluating the $g(2)$ from figure 3 should give a

more reliable value for the reader. A longer integration time could have been helpful to increase the signal to noise ratio. Finally, in figure 3 and S5, coincidences are not normalized, making difficult to understand the results.

6) In S2, it is said that coincidences where the SAW current drifts above $1e_f$ and below $0.8e_f$ are removed. Does this mean that the $g(2)$ of 0.39 is post selected? If so, the authors should mark it clearly in the manuscript and say which is the measured $g(2)$ as raw data: this is an important number in the experiments.

7) A strong point is made that this approach is more reliable for scaling up the device fabrication but the operation temperature is 1.5K (much more difficult to achieve than the usual 4K used for defect centres or quantum dots). How the performances degrade if the system is operated at higher temperatures?

8) How is it possible to make sure the exciton recombination does not happen directly below the metal contact (this would limit the photon extraction)?

9) How the 2.5% collection is estimated? The authors should provide more information.

10) It is stated that no filtering is required for this measurement. On the other hand, in the methods section it is stated that a spectrometer was used. For the Hanbury Brown-Twiss were the SPAD placed after or before the spectrometer?

Despite in the current form the manuscript does not seem to compare fairly with other systems, I do believe the results are of high quality and deserve publication.

Minor:

It could be useful to mark in Fig. 1 where the p-i-n regions are. This would improve the clarity.

Reply to the reviewers' comments

Dear Editor and reviewers,

We thank you for your time and careful consideration of our manuscript. This document contains the reviewers' comments for our manuscript, and our reply. The manuscript has been revised according to both reviewers' comments and suggestions. Modified paragraphs in the paper are highlighted in red.

Yours sincerely,
Tzu-Kan Hsiao and Chris Ford.

Response to referees

Reviewer #1:

1) The authors report on an unconventional type of single photon source. This device consist of an electrostatically induced p-n junction in a two-dimensional quantum well.

Following a proposal dating back to 2000, electrons transported one-by-one via the acoustic-electrical effect into the p-type region, a single photon is emitted. Single photon emission is verified by performing Hanbury Brown & Twiss intensity interferometry. the observed suppression of correlations at time=0 demonstrates non-classical light emission.

The $g^{(2)}(0)$ value is slightly below 0.5, which is the limit for a perfect source emitting in a single optical mode. This overall performance of the device is still limited but the authors argue that it can be significantly improved by incorporating the structure into an optical cavity to extract light. This reviewer agrees with the authors that this may in fact be feasible.

This reviewer considers the reported research as important to the field of semiconductor quantum electronics because it provides an interface to convert chip-based electron qubits to flying photonic qubits. This reviewer believes that this may be the main application of the demonstrated paradigm and NOT single photon emission. Therefore, this reviewer expects broader impact also in the field of quantum transport and spin-based quantum computing.

This reviewer also recognises that the authors report the first demonstration of this particular type of single photon source.

This reviewer considers the presented work as correct and of high scientific rigour.

The points of novelty listed above warrant in this reviewer's opinion publication in Nature Communications.

Before this reviewer can accept publication, the authors are requested to consider the following two points.

We thank reviewer 1 for recognising the novelty and application potential in our work.

2) In their manuscript and the motivation of the work the authors set a very strong focus on single photon emission. This field is very advanced, in particular extremely bright, electrically driven sources comprising quantum dots and microresonators are commercially available. Similar holds for room temperature compatible sources using nitrogen vacancy centres in diamond. This reviewer is convinced that the source reported here will not be competitive but find different applications.

In this context this reviewer recommends to improve the manuscript: The authors cite a long list of works on different types of single photon source (various types of quantum dots, molecules, PDC), several review papers (some on exactly the same topic) and proposals (Quantum internet, linear QCs, etc.) employing single photons.

Work on acoustically driven sources is not made as prominent as it should be. In particular there exists an alternative scheme using quantum dots and the acoustoelectric effect of which only the very first realisation is referenced (ref. 32). There is also a significant body of literature studying the manipulation of optically active quantum emitters, in particular quantum dots via the dynamic strain field of an acoustic wave and regulating the carrier injection by the acoustoelectric effect. This reviewer is convinced the authors are aware of these works and recommends a careful revision such that the reported important results are put in the full context of the field.

We have revised our manuscript to more emphasise its impact on acoustoelectric transport and quantum computing.

3) The authors mention that the coherence length of their device has to be improved to meet the requirements for their envisioned applications in quantum repeaters. What is the coherence length of the photons emitted by their current device?

The FWHM of 1 meV in our current device corresponds to a coherence time of 1 ps, hence a coherence length of $300 \mu\text{m}$. By using a quantum-well with better interface roughness, the coherence length may be increased to 3.7 mm (as discussed in the supplementary information).

Reviewer #2:

4) The authors report on the generation of single photons by injecting (ideally) single carriers using a surface acoustic waves. The full device is fabricated using standard lithography techniques which makes it appealing in terms of reproducibility. The method is interesting, experiments quite well conducted and results are generally

well described. I think that the manuscript deserves publication in Nature Communications. Although, there are some major modifications needed before final acceptance.

As a general statement, the results are quite impressive but the authors keep referring to applications and performances they did not investigate (and probably very hard to reach in the future). Since generation of single photons was proven, they should compare with results achieved using electrically driven sources (on various platforms). In several points of the manuscript, they try to convince the reader that their device can generate indistinguishable photons or entangled photons better than other structures (in particular referring to quantum dots). This does not highlight the nice achieved results but forces the reader to compare highly developed performances with arguable theoretical assumptions. This deviates the attention from current data. I suggest the authors to rephrase introduction, conclusion and other parts of the manuscript accordingly. In more details:

We are pleased that the reviewer recommends publication in Nature Communications and we thank the reviewer for the helpful suggestions. We have modified our manuscript accordingly.

5) Quantum dots under optical pumping demonstrated very high degree of indistinguishability (see Ref. [14, 15, 16]) and high extraction efficiency. Additionally, for quantum dots with a lifetime of around 1 ns, linewidth ≤ 1 GHz were reported. The current device generates photons with a bandwidth of 1 meV: the reported lifetime of 100 ps is said to be limited by non-radiative processes; in any case this value is still far from the Fourier limit. This would have a dramatic effect in multiphoton experiments (see H. Wang, et al, Phys. Rev. Lett. 116, 213601 (2016) or J. C. Loredó, et al, Optica 3, 433-440 (2016)).

The single-photon purity, indistinguishability and extraction efficiency of our current device are indeed not as good as for mature quantum-dot single-photon sources. We hope that the single-photon performance based on our new method will some day be improved using strategies outlined in our manuscript and supplementary information. However, we admit that the photon indistinguishability from our scheme may not compete with self-assembled quantum dots, but may be more suitable for integrated on-chip photonic device or electron-to-photon qubit conversion.

6) The current results compare very well with electrically driven sources and they would have much better impact if discussed accordingly (see few examples: T. Heindel, et al. New J. Phys. 14, 083001 (2012), J. Zhang, et al, Nat. Commun. 6, 10067 (2015), M. Sartison, et al, Appl. Phys. Lett. 114, 222101 (2019), Z. Yuan, et al, Science 295, 102 (2002), T. Müller, et al, Nat. Commun. 9, 862 (2018))

Thanks for the suggestion. We have added more discussion on electrically-driven sources in the first paragraph.

7) Since the current device produces single photons with a potentially high fabrication yield, I suggest the authors rather than mentioning "Quantum communication and quantum computation", to target quantum cryptography (where only single photons are required).

In a relatively short range (like within a city or a small country), quantum cryptography does require only high single-photon purity and high repetition rate. However, for long-range quantum cryptography, quantum repeaters will be necessary to extend communication distance. In that case, photon indistinguishability will also be as vital as in quantum computation.

8) In a similar argument, the generation of polarization entangled photons is well established with quantum dots (see D. Huber, et al, Phys. Rev. Lett. 121, 033902 (2018) or Y. Chen, et al, Nat. Commun. 9, 2994 (2018)). The current device does not need to show this capability and the proposed scheme (polarisation-to-spin conversion) seems quite difficult to implement.

Indeed, a lot of effort will be needed to investigate if our method can be used to generate entangled photons, which is beyond the scope of our current work, though in principle, pairs of pumped electrons will already be entangled if in their ground state, and the difficulty will be getting them to create photons with a constant delay between them. About polarisation-to-spin conversion, a very recent paper has just demonstrated such a technology, showing that the proposed scheme may be difficult but should be feasible [1].

9) In the evaluation of the $g(2)$ function, something seems unclear. The device is operated under pulsed excitation, so that a clear stream of individual peaks can be observed in figure 3 and figure S5. Then a complex procedure to estimate the $g(2)$ is used, obtaining a $g(2)$ function as in figure 4 which looks like a CW second order correlation function. Which new information is given by this figure? Why the authors have to process the data in this way? It is claimed for a better accuracy but adding complex fitting routines should increase the uncertainty. Indeed, the peaks are well separated (despite a slight overlap) and the fitting of figure S5 looks fine. Nevertheless no value of $g(2)$ is reported, while it should be around 0.39. Evaluating the $g(2)$ from figure 3 should give a more reliable value for the reader. A longer integration time could have been helpful to increase the signal to noise ratio. Finally, in figure 3 and S5, coincidences are not normalized, making difficult to understand the results.

The main reasons we obtain $g^{(2)}(0)$ using this analysis process are:

(1) we know there is a non-zero background in the EL signal (Fig. 2(b)), possibly due to after-pulsing of the SPAD or to slowly-decaying secondary-exciton states (lifetime $\sim 0.2 - 1.5$ ns). This EL background can lead to a background in the autocorrelation histogram (Fig. 3(a) and Fig. S5), which will make the peak at $\Delta t = 0$ higher.

(2) As reviewer 2 pointed out, there is overlap between two neighbouring peaks in Fig. 3(a) and Fig. S5. This overlap can also affect the height of the peak at $\Delta t = 0$.

(3) The line-shape is constant and the fit to each peak is good. This enables us to use the full amount of data in each period to reduce noise, rather than just integrating across the period, which includes the background and the tail of the previous peak, as described above.

Hence, in Fig. S5, we obtain the amplitudes of each peak by considering the effect from the EL background and the shape of each peak. We then plot the amplitudes of 100 peaks around $\Delta t = 0$ in Fig. 4 so that the background and the effect of overlap can be eliminated. If we simply determine $g^{(2)}(0)$ from the peak height in Fig. 3(a), the value is about 0.59 although this value is not representative because of the background and peak overlap. We mention this number in the revised manuscript and normalise the coincidences in Fig. 3(a).

10) In S2, it is said that coincidences where the SAW current drifts above 1ef and below 0.8ef are removed. Does this mean that the $g^{(2)}$ of 0.39 is post selected? If so, the authors should mark it clearly in the manuscript and say which is the measured $g^{(2)}$ as raw data: this is an important number in the experiments.

Yes, $g^{(2)}(0)$ is in a sense post-selected since we know that outside this range the device is not in the single-electron-transport regime. Before removing coincidences outside this regime, $g^{(2)}(0)$ is 0.63 ± 0.04 (0.78 ± 0.05 if simply determined from peak height in raw data). We describe this more clearly in the revised manuscript.

11) A strong point is made that this approach is more reliable for scaling up the device fabrication but the operation temperature is 1.5K (much more difficult to achieve than the usual 4K used for defect centres or quantum dots). How the performances degrade if the system is operated at higher temperatures?

We did not test our device at 4K. However, the addition energy of the SAW confinement is estimated to be $2.6 \text{ meV} = 30\text{K} \times k_B$ at the power of 11 dBm in our previous work [2]. The difference in thermal energy between 1.5 K and 4 K is $\sim 0.2 \text{ meV}$, which is small compared with the addition energy. Hence, we believe that working at 4 K may not impact the single-electron-transport probability and the corresponding single-photon purity ($g^{(2)}(0)$). Higher temperature will also broaden the FWHM of EL spectrum, therefore degrade the photon indistinguishability.

12) How is it possible to make sure the exciton recombination does not happen directly below the metal contact (this would limit the photon extraction)?

We think now there is actually some recombination happening below the hole-inducing surface gate, which contributes to the low quantum efficiency. The Ti/Au surface-gate thickness is 5/10 nm, which should have optical transmission of 40% [3]. Making the gate thinner can improve the transparency. Another option is to use transparent conductive oxide like indium-tin-oxide (ITO) thin films, which have a high transparency of $\sim 90\%$. On the other hand, one can also make the voltage on the gate more negative in order to stop SAW-driven electrons from going further.

13) How the 2.5% collection is estimated? The authors should provide more information.

The internal quantum efficiency η , defined as the ratio of the number of photons actually collected to the number of photons that can theoretically be collected by the optics. In the single-electron-transport regime, η can be determined by

$$\eta = \frac{N_{detect}}{f_{SAW} \cdot C_{optics} \cdot C_{SPAD}} \quad (1)$$

where N_{detect} is the number of photons detected by the single-photon avalanche photodiode (SPAD) per second, f_{SAW} is the SAW frequency (corresponding to the number of recombination per second), C_{optics} is the collection efficiency of the lens assembly ($\sim 0.4\%$), and C_{SPAD} is the sensitivity of the SPAD at 800 nm ($\sim 15\%$). We have added the information about collection efficiency in the Methods section.

14) It is stated that no filtering is required for this measurement. On the other hand, in the methods section it is stated that a spectrometer was used. For the Hanbury Brown-Twiss were the SPAD placed after or before the spectrometer?

The spectrometer was used for measuring the EL spectrum, as shown in Fig. 1(e). There was no spectrometer or any spectral filtering involved when we took the HBT histogram. We now make this clear in the Methods section.

15) Despite in the current form the manuscript does not seem to compare fairly with other systems, I do believe the results are of high quality and deserve publication.

Minor:

It could be useful to mark in Fig. 1 where the p-i-n regions are. This would improve the clarity.

We have made it clearer in the figure caption. Thanks for the suggestion.

References

- [1] Fujita, T. *et al.* Angular momentum transfer from photon polarization to an electron spin in a gate-defined quantum dot. *Nature Communications* **10**, 2991 (2019).
- [2] Astley, M. R. *et al.* Energy-dependent tunneling from few-electron dynamic quantum dots. *Physical Review Letters* **99**, 156802 (2007). 0708.3746.
- [3] Leosson, K. *et al.* Ultra-thin gold films on transparent polymers. *Nanophotonics* (2013).

Reviewers' Comments:

Reviewer #1:

Remarks to the Author:

The authors revised the manuscript taking into account the comments raised by both reviewers.

I acknowledge the efforts made by the authors. However, I still have reservations and cannot publication in the present form.

The work reported is very nice and deserves publication in Nature Communications. My main concern is still the framing of the work. I am convinced (and from the report of the other reviewer I feel strongly confirm in this view) that the demonstrated structure will not find applications as a single photon source. The main point of novelty is in my view bridging between two realms in SAW research, single carrier (spin/charge) transport and optics. This is indeed in line with the criteria of Nature Communications: "Papers published by the journal represent important advances of significance to specialists within each field."

There's definitely a long way to go to make this system efficient and deterministic conversion of a spin qubit to a polarization qubit of the single photon. The reported work is clearly the first and very important step towards this ambitious goal.

In the present manuscript all data analysis are made such to show that there is non-classical light emission. There is apparently a lot of processing (postselection etc) done solely for the purpose to get towards performance levels compatible with a single photon source. The $g(2) \leq 0.5$ criteria applies for instance for a perfect emitter coupled to a single optical mode and only the most advanced devices may get close to this.

Based on what I written above, I am more than happy support publication, but only in the case that the authors really take the time and frame their work differently. It should be placed primarily in the context of charge/spin to photon conversion in the single carrier limit. I am more than happy if the single photon source aspect is included but it should not be the main emphasis. One final note: I appreciate that the authors will make data available according to the Data Availability statement in the manuscript. I encourage them to include all raw data and sufficient details on the data analysis.

===== End report =====

Reviewer #2:

Remarks to the Author:

I appreciate the efforts of the authors in modifying the manuscript but, in this new version, a serious problem appeared.

The major novelty point of this study was said to be the proof of single photon emission from this SAW-driven device. On the other hand, the authors did not mention in their manuscript (also after revision) that the actual measured $g(2)$ is clearly above 0.5. They give an explanation of the non-single photon emission, even without giving proper number (like, how many coincidences have been removed?), but they do not want to discuss how this impacts the device performances. It is true that the light emission is "single" for some time, nevertheless it is clearly "not single" for a considerable amount ($g(2)$ of 0.63 without removing coincidences).

While I still appreciate the work of the authors in realizing this device, performing the experiments and analyze the data, I do not think that the degree of novelty is enough to grant publication in Nature Communications, where clear steps forward must be achieved. In the present work, sub-

Poissonian photon emission is proven but the oscillations in the current prevent the generation of actual single photons (0.63 is the number that matters in applications).

I would recommend the authors to resubmit this work in a more specialized journal. Also, I suggest to explicitly add the discussion on the $g(2)$ in the manuscript, reporting its non post-selected value and the respective values in the probability estimation. Only in this way the manuscript can be useful for the scientific community that can compare these performances to their results.

While the current results are not enough to grant a publication in Nature Communications, a sound explanation on the limiting factors and how to achieve non-post selected $g(2) < 0.5$ can be of interest to a more specialized audience.

Reply to the reviewers' comments

Dear Reviewers,

We thank you for your time and feedback for our manuscript. We have revised the paper based on reviewers' suggestions. Please see our reply to the comments below. We hope you will now find our revised manuscript clear and ready to be published in *Nature Communications*.

Yours sincerely,
Tzu-Kan Hsiao and Chris Ford.

Response to reviewers

Reviewer #1:

1) The authors revised the manuscript taking into account the comments raised by both reviewers.

I acknowledge the efforts made by the authors. However, I still have reservations and cannot publication in the present form. The work reported is very nice and deserves publication in *Nature Communications*. My main concern is still the framing of the work. I am convinced (and from the report of the other reviewer I feel strongly confirm in this view) that the demonstrated structure will not find applications as a single photon source. The main point of novelty is in my view bridging between two realms in SAW research, single carrier (spin/charge) transport and optics. This is indeed in line with with criteria of *Nature Communications*: Papers published by the journal represent important advances of significance to specialists within each field.

There's definitely a long way to go to make this system efficient and deterministic conversion of a spin qubit to a polarization qubit of the single photon. The reported work is clearly the first and very important step towards this ambitious goal.

We thank reviewer 1 for the suggestion of reframing our work to increase its impact in the quantum-information community. In the revised manuscript we put our main focus on realising the first major step toward electron-to-photon qubit conversion. We still mention single-photon applications in the supplementary information to relate our work to Foden's scheme proposed 20 years ago.

2) In the present manuscript all data analysis are made such to show that there is non-classical light emission. There is apparently a lot of processing (postselection etc) done solely for the purpose to get towards performance levels compatible with a single photon source. The $g(2) \leq 0.5$ criteria applies for instance for a perfect

emitter coupled to a single optical mode and only the most advanced devices may get close to this.

Based on what I written above, I am more than happy support publication, but only in the case that the authors really take the time and frame their work differently. It should be placed primarily in the context of charge/spin to photon conversion in the single carrier limit. I am more than happy if the single photon source aspect is included but it should not be the main emphasis.

One final note: I appreciate that the authors will make data available according to the Data Availability statement in the manuscript. I encourage them to include all raw data and sufficient details on the data analysis.

We thank reviewer 1 again for supporting the publication in Nature Communications. We have reframed our work in the context of electron-to-photon conversion in the single-carrier limit. We will also upload the raw data and our scripts for analysis in Cambridge data repository (<https://www.repository.cam.ac.uk/>).

Reviewer #2:

3) I appreciate the efforts of the authors in modifying the manuscript but, in this new version, a serious problem appeared.

The major novelty point of this study was said to be the proof of single photon emission from this SAW-driven device. On the other hand, the authors did not mention in their manuscript (also after revision) that the actual measured $g(2)$ is clearly above 0.5. They give an explanation of the non-single photon emission, even without giving proper number (like, how many coincidences have been removed?), but they do not want to discuss how this impacts the device performances. It this true that the light emission is 'single' for some time, nevertheless it is clearly 'not single' for a considerably amount ($g(2)$ of 0.63 without removing coincidences).

We thank reviewer 2 for the feedback for the revised manuscript. The main focus of our work is that we have achieved photon antibunching and single-photon emission when the SAW transport is in the single-electron regime. That is why we want to report $g^2(0)$ when the SAW-driven current is between $0.8ef$ and $1ef$ (multi-electron and vacuum states both lead to $g^2(0) \geq 0.5$).

One issue of our prototypical device is that the SAW-driven current drifts away from the single-electron regime, possibly due to charging of surface states near the etched 1D channel. Also, as we mentioned in the manuscript, the internal quantum efficiency is quite low ($\sim 2.5\%$). Combined with the collection efficiency of the lens assembly (0.4%) and the SPAD sensitivity (15%), this low overall detection efficiency results in a long measurement time of 2-3 days in order to have enough coincidences. Even though we use a feedback loop to stabilise the current, it can still drift outside the desired single-electron regime when the current drifts faster than the feedback. Therefore, we remove coincidences in the multi-electron and vacuum regimes, and keep the 30% of coincidences that occurred in the single-electron regime.

It is not our intention to hide the raw $g^2(0) = 0.63$. We did mention this number

in the reply to reviewer 2 but should also have made it clear in the main text. In the revised manuscript we explicitly report this number in the abstract and the main text. In the supplementary information we also present the data analysis and photon-number probability for the raw data, showing that even without post-selection, for the continuous duration of 2-3 days, we can still observe clear photon-antibunching and enhanced single-photon probability compared with a classical Poissonian light source. Interestingly, this post-selection can be done simultaneously in the measurement because we can know if the device is in the single-electron regime or not by measuring the current in real time rather than post-selecting by looking for photon correlations.

Currently the performance of our prototypical device is limited by the charging effect and the low quantum efficiency. As we discussed in the main text and supplementary information, non-radiative surface states may be reduced by surface passivation to improve the device stability and quantum efficiency. Building a Bragg stack to form an optical cavity, and introducing extra electrostatic confinement near the hole region, will also increase the efficiency. With these improvements, one can measure the HBT histogram in a shorter time and with less current drift, so that our current result $g^2(0) = 0.39$ or better may be achievable without post-selection. To further improve $g^2(0)$, as we already discussed in the main text, one can use a narrower channel for the lateral confinement and stronger SAW potential for the longitudinal confinement to increase the charging energy of the dynamic quantum dot.

4) While I still appreciate the work of the authors in realizing this device, performing the experiments and analyze the data, I do not think that the degree of novelty is enough to grant publication in Nature Communications, where clear steps forward must be achieved. In the present work, sub-Poissonian photon emission is proven but the oscillations in the current prevent the generation of actual single photons (0.63 is the number that matters in applications).

I would recommend the authors to resubmit this work in a more specialized journal. Also, I suggest to explicitly add the discussion on the $g(2)$ in the manuscript, reporting its non post-selected value and the respective values in the probability estimation. Only in this way the manuscript can be useful for the scientific community that can compare these performances to their results.

While the current results are not enough to grant a publication in Nature Communications, a sound explanation on the limiting factors and how to achieve non-post selected $g(2) \leq 0.5$ can be of interest to a more specialized audience.

As answered in the paragraph above, we have added more discussion about $g^2(0)$ for the raw data. We hope reviewer 2 will find our result of realising $g^2(0) \leq 0.5$ in the single-electron-transport regime clear and support its publication in Nature Communications.